# Factors Associated with a Post-Procedure Spontaneous Pregnancy after a Hysterosapingo-Foam-Sonography (HyFoSy): Results from a Multicenter Observational Study

**DOI:** 10.3390/diagnostics13030504

**Published:** 2023-01-30

**Authors:** Virginia Engels, Margarita Medina, Eugenia Antolín, Cristina Ros, Carmina Bermejo, Nabil Manzour, Irene Pelayo, Ainara Amaro, Pilar Martinez-Ten, Cristian De-Guirior, Roberto Rodríguez, Laura Sotillo, Isabel Brotons, Reyes de la Cuesta-Benjumea, Oscar Martinez, Javier Sancho, Juan Luis Alcázar

**Affiliations:** 1Department of Obstetrics and Gynecology, Hospital Universitario Puerta de Hierro, 28222 Majadahonda, Spain; 2Department of Obstetrics and Gynecology, Hospital Universitario Materno-Infantil, Las Palmas de Gran, 35016 Canaria, Spain; 3Division of Maternal and Fetal Medicine, Department of Obstetrics and Gynecology, Hospital Universitario La Paz. IdiPAZ, 28046 Madrid, Spain; 4Department of Gynecology, Institut Clínic de Ginecologia, Obstetrícia, i Neonatologia, Hospital Clinic of Barcelona, University of Barcelona, 08036 Barcelona, Spain; 5Delta Ecografía, Centro de Diagnóstico por la Imagen en Obstetricia, Ginecología y Mama, 28006 Madrid, Spain; 6Department of Obstetrics and Gynecology, Clinica Universidad de Navarra, 31008 Pamplona, Spain; 7Department of Obstetrics and Gynecology, Hospital Universitario Ramón y Cajal, 28034 Madrid, Spain

**Keywords:** hysterosalpingo-foam-sonography (HyFoSy), infertility, spontaneous conception, pregnancy, tubal flushing effect

## Abstract

Background: Tubal patency testing constitutes an essential part of infertility work-up. Hysterosalpingo-foam-sonography (HyFoSy) is currently one of the best tests for assessing tubal patency. The objective of our study was to evaluate the post-procedure rate of spontaneous pregnancy among infertile women submitted for an HyFoSy exam with ExEm^®^ foam and the factors associated with this. Methods: Multicenter, prospective, observational study performed at six Spanish centers for gynecologic sonography and human reproduction. From December 2015 to June 2021, 799 infertile women underwent HyFoSy registration consecutively. The patients’ information was collected from their medical records. Multivariable regression analyses were performed, controlling for age, etiology, and time of sterility. The main outcome was to measure post-procedure spontaneous pregnancy rates and the factors associated with the achievement of pregnancy. Results: 201 (26.5%) women got spontaneous conception (SC group), whereas 557 (73.5%) women did not get pregnant (non-spontaneous conception group, NSC). The median time for reaching SC after HyFoSy was 4 months (CI 95% 3.1–4.9), 18.9% of them occurring the same month of the procedure. Couples with less than 18 months of infertility were 93% more likely to get pregnant after HyFoSy (OR 1.93, 95% CI 1.34–2.81; *p* < 0.001); SC were two times more frequent in women under 35 years with unexplained infertility (OR 2.22, 95% CI 1.07–4.65; P0.033). Conclusion: After HyFoSy, one in four patients got pregnant within the next twelve months. Couples with shorter infertility time, unexplained infertility, and women under 35 years are more likely to achieve SC after HyFoSy.

## 1. Introduction

Infertility is the inability to conceive after having regular unprotected sexual intercourse for 12 months [1]. This problem eventually affects 8–12% of couples of reproductive age and is increasing due to the delay in searching for the first pregnancy [2]. The tubal factor accounts for about 11–67% of infertility diagnoses [3]. Laparoscopy with a dye test is considered the gold-standard method for assessing tubal patency. However, the invasiveness of this procedure precludes its inclusion in most infertility guidelines. 

Traditionally, an X-ray hysterosalpingogram (HSG) has been the first-line test for tubal patency assessment. Nevertheless, this technique has some disadvantages, such as being painful, employing iodinated contrast media, and delivering ionizing radiation [4,5]. Recent, new outpatient techniques have been proposed as alternatives to classical HSG, such as hysterosalpingo-foam-sonography (HyFoSy), which is a less painful alternative to HSG consisting of an intrauterine ultrasound contrast media instillation followed by a transvaginal ultrasound scan [6] and selective chromopertubation via office hysteroscopy [7]. The ExEm^®^ gel and foam is a specifically developed contrast for gynecological use, safe for intrauterine application, and specifically designed for tubal patency ultrasonographic diagnosis [6]. Performed by expert gynecologists, HyFoSy presents clear advantages over HSG, since it is slightly invasive and avoids the risk of iodinate contrasts and ionizing radiation exposure [6]. Moreover, the reliability and accuracy of HyFoSy [8,9] have been tested with very good results, as did tolerability, feasibility, and safety [10,11]. Finally, HyFoSy enables a one-step examination in the evaluation of the endometrial cavity and the detection of gynecological pathology. Therefore, HyFoSy is becoming a first-line tool, which is displacing the former HSG [12]. 

X-ray HSG is associated with significant rates of spontaneous pregnancies after the procedure, mostly while oil-soluble contrast media are employed [13]. The potential therapeutic effect (tubal flushing effect) of HSG represents a clear advantage of the technique that is appreciated by both infertile patients and gynecologists. Regarding the HyFoSy test, some recent studies have been reported suggesting the possibility of a potential “tubal flushing effect”, enhancing the chance of post-procedure spontaneous pregnancy within a few months after the procedure. Tanaka and co-workers observed that 46% of women who underwent HyFoSy got pregnant spontaneously within 6 months after the procedure as compared to 41% of those who underwent assisted reproductive techniques [14]. On the other hand, Lindborg and colleagues found that the pregnancy rate after HyFoSy was 29% in a randomized controlled trial [15].

Nevertheless, there is a lack of strong evidence for this phenomenon, for most of the studies are retrospective [14,16], include few patients [17], or evaluate tests with contrasts other than ExEm^®^ foam [18,19]. On the other hand, there are few data about which factors could be associated with post-procedure spontaneous pregnancy. Therefore, the aim of the study was to evaluate prospectively the post-procedure spontaneous conception (SC) rate in a large multicenter study, including infertile women submitted to HyFoSy using ExEm^®^ foam contrast and analyzing what factors are associated with this outcome.

## 2. Materials and Methods

### 2.1. Patients 

The study was conducted prospectively from December 2015 to June 2016 in six Spanish university teaching hospitals. The Institutional Ethics Committee of each center approved the study protocol. Infertile women scheduled for HyFoSy for tubal patency testing as a part of the fertility work-up were invited to participate in the study. Investigators explained the purpose of the study, and if women agreed, they signed a written consent for their participation.

Inclusion criteria were as follows: nulliparous women aged over 18 years with at least 12 months of infertility and a candidate for ovarian stimulation (OS) and intrauterine insemination (IUI) [unexplained infertility, mild-moderate male factor infertility (WHO [20]) and ovarian dysfunction]. Exclusion criteria were as follows: secondary infertility, severe male factor, previous diagnosis of endometriosis, women candidate for donor semen IUI, or women with an SC of more than 12 months after HyfoSy. 

After HyFoSy, patients were informed of their tubal patency status, giving them the opportunity to be treated with a maximum of four consecutive IUIs when at least one tube was patent. Bilateral tubal occlusion on the HyFoSy test was an exclusion criterion for IUI, but not for the inclusion in this study; women with bilateral occlusion on HyFoSy could achieve spontaneous pregnancy since previous authors report false tubal occlusions on HyFoSy scans [21].

### 2.2. Interventions

HyFoSy procedures were scheduled in the late follicular period (days 7–11 of the menstrual period). Women took antibiotic prophylaxis with 1 g of oral azithromycin the night before the exam. All women were examined in the lithotomy position. The external cervical os was visualized using a vaginal speculum and cleaned with a 3% iodine povidone solution. Thereafter, the endocervix was cannulated with either a pediatric nasogastric tube, an intrauterine insemination catheter, or a specifically designed HyFoSy catheter. In very unfavorable cases, cervices were gripped with reusable Pozzi forceps to favor the entrance. 

The foam was prepared following the instructions of the manufacturer (IQ Medical Ventures BV, Rotterdam, The Netherlands). Gynecologists with extensive experience in gynecological ultrasound (all examiners had more than 15 years’ experience in gynecological ultrasound scanning) using high-end equipment (Voluson 730 Expert, Voluson E8 and Voluson E10 [GE Medical Systems, Zipf, Austria]) performed all the scans using the same protocol. Briefly, the uterus was scanned from horn to horn in the sagittal plane and from the cervix to the fundus in the transverse plane. The ultrasound image was magnified to contain the uterine corpus and cervix. The sonographic image focused on the tip of the catheter. The foam was slowly injected (one mL by one mL) via the catheter to assess the patency of the fallopian tubes. The sonographic probe was moved from horn to horn, and the fallopian tubes were evaluated for 5–7 minutes since the foam was stable and showed echogenicity during this lapse of time. Tubal patency was established by the visualization of the echogenic contrast advancing through the tubes, flushing through the fimbrial part of the tube, and expanding around the ovaries. 

### 2.3. Study Outcomes

Spontaneous conception (SC) achieved within 365 days (12 months) after HyFoSy was considered the main outcome of the study. Therefore, patients were divided into two groups; group A: patients who achieved SC after HyFoSy (SC group) and group B: patients who did not achieve SC HyFoSy (Non-SC group). We decided to consider 12 months post procedure since sterility is defined as “the inability to attend conception after 12 months of unprotected sexual intercourse” [1]. 

Even if the probability of SC was the primary endpoint of the study, patients were offered to initiate, without delay, IUI cycles. Therefore, SC could happen before starting IUI or in the intervals between cycles (holiday periods, temporary contraindication to perform IUI due to COVID infection, persistent follicle in baseline sonography, or personal reasons of the couples). The main reason to go on with the IUI procedures, even if SCs were the endpoint of the study, was the loss of opportunity that could suppose a delay in the treatment of infertility that could lead to a decay in the chances of pregnancy. Therefore, in the Non-SC group, we included patients with no conception at all during the study period and those who reached it through uterine insemination or in vitro fertilization techniques after the HyFoSy procedure. 

### 2.4. Variables and Statistical Analysis

Information regarding the baseline characteristics and fertility parameters was extracted from the clinical history of each participant. Data regarding the procedure and subsequent events were recorded prospectively during the participant’s follow-up period. The following variables were analyzed: patient’s age, patient’s body mass index (BMI), months attempting to get pregnant (months of infertility), cause of infertility (ovulatory, mild or moderate male factor, tubal factor, uterine factor, endocrine factors, mixed causes or unexplained) and HyFoSy parameters (contrast volume –in mL-, pain associated (Visual analog scale) and presence of unilateral or bilateral tubal occlusion). 

It is well known that fertility decreases significantly in women older than 35 years [22]. Therefore, we decided to dichotomize the patients’ ages into two groups: ≥35 years old and <35 years old. On the other hand, infertility duration is also a factor related to treatment outcome [23]; thus, the variable “months of infertility” was also dichotomized into two groups: ≥18 months and <18 months. Finally, as the cause of infertility is another factor related to treatment outcome [23], we dichotomized the variable “cause of infertility” into two groups: “unexplained fertility” and “explained infertility”.

The numerical variables were tested for normal distribution using the Kolmogorov-Smirnov test. Descriptive data are presented as mean with standard deviation or median with interquartile range (IQR) for numerical variables, depending on normal distribution and number and percentage for categorical variables. *p*-values of the univariable analysis were obtained by one-way ANOVA or Mann-Whitney´s U tests for numerical and Pearson chi-squared or Fisher´s exact test for categorical variables. In addition, time to SC after HyFoSy was analyzed by Kaplan–Meier curves and the log-rank test. Statistical tests were two-sided and were performed with SPSS V.20 [IBM Inc., Chicago, IL, USA]; a *p*-value below 0.05 was considered significant. 

Multivariable logistic regression modeling was conducted to derive the adjusted odds ratio [aOR] with a 95% confidence interval [95% CI] of “SC risks factors” found in the previous univariable analysis. The regression analysis was carried out using the Ime4 package in R, version 3.4 (RCoreTeam, 2017) [24].

## 3. Results

### 3.1. General Data 

We performed an HyFoSy examination on 799 women, one test per woman. Two women were excluded because of a failure in the cervical cannulation (failure rate 0.25%). In 37 women, the contrast used was other than ExEm^®^ foam, and they were excluded from the final analyses. Two patients were lost to follow-up and information about SC could not be obtained from these patients; they were also excluded. Therefore, 758 patients were ultimately included in the study. Of these 758 patients, 201 (26.5%) became spontaneously pregnant (SC group) during the 12 months of follow-up, while 557 (73.5%) did not conceive spontaneously (Non-SC group) (Figure 1).

The median time for reaching pregnancy among SC was 4 months (CI 95% 3.1–4.9), with 18.9% of the pregnancies occurring in the month after the procedure (Figure 2).

### 3.2. Patients’ Characteristics and Fertility Parameters 

The median age of the SC group was 34 years [IQR 30–36], and the median body mass index was 23.0 (IQR 20.0–27.0). We observed that patients aged below 35 years tended to get pregnant after HyFoSy more frequently than those aged over 35 years (Table 1).

Causes of infertility in both SC and non-SC groups were ovulatory dysfunction (10.4% vs. 20.0%), unilateral tubal occlusion (10.9% vs. 6.8%), mild or moderate male factor (15.8% vs. 18.1%), mixed sterility factors (7.7% vs. 13.0%), and unexplained infertility (42.1% vs. 32.2%), respectively. When grouping this variable as “unexplained infertility” and “explained infertility”, women with SC had unexplained infertility more frequently than those in the Non-SC group (*p* = 0.017). 

The median duration of infertility was 18 months (IQR 12–24) in both groups. However, we did observe that 69.1% of the patients who got SC after HyFoSy had an infertility duration of less than 18 months compared to 54.9% in the Non-SC group (*p* = 0.0006).

### 3.3. Characteristics Related to HyFoSy Procedure and Findings 

During the exams, the median volume of foam instilled was 5 mL in both groups (Table 2). No significant difference was observed. Regarding tubal patency, both fallopian tubes were patent in 76.6% of SC patients and 76.2% of Non-SC groups. Unilateral patency was observed in 18.9% and 19.5% in the SC group and Non-SC groups, respectively. Bilateral fallopian tube occlusion was observed in 4.5% and 4.2% of the patients in SC and Non-SC groups, respectively. No significant difference was found between both groups for this parameter.

Median pain during the procedure on the VAS scale was 2 (IQR 1–5) in Non-SC and 3 (IQR 1–4) in SC, *p* = 0.634), which translates into two-thirds of the patients in both groups referring mild pain (Table 2). No severe complications were reported in this series. Only two patients had vagal syndrome, and one patient had a mild urinary infection. 

### 3.4. Multivariable Analysis for SC after HyFoSy Examination

We considered for the multivariate analysis those variables found to be statistically significant in the univariate analysis. Namely, age group (<35 years old versus ≥35 years old), time of infertility (<18 months versus ≥18 months), and cause of infertility (unexplained vs. explained).

After multivariate analysis, we found that couples with less than 18 infertility months were 93% more likely to get pregnant after HyFoSy compared to those with more than 18 months of proven sterility. In addition, SC after HyFoSy was 2 times more likely to happen in women aged under 35 years who were diagnosed with unexplained infertility (Table 3).

### 3.5. Perinatal Outcomes of SC after the Procedure 

When the perinatal outcomes were compared between those pregnancies achieved after HyFoSy alone versus those achieved after HyFoSy followed by IUI, no statistical differences were observed in terms of miscarriage rate, preterm births, and fetal malformations. However, the weight at birth was significantly higher in neonates born after HyFoSy alone (*p* = 0.034) (Table 4).

## 4. Discussion

### 4.1. Summary of Findings

The main finding of our study is that 26.5% of patients became spontaneously pregnant during the 12 months of follow-up after HyFoSy. Couples with an infertility duration of less than 18 months and women under 35 years with unexplained infertility were more likely to get pregnant after HyFoSy.

### 4.2. Interpretation of Results

The potential relationship between tubal patency tests and subsequent spontaneous pregnancy has been acknowledged for more than fifty years [25]. A recent systematic review analyzing data from 15 clinical trials found an OR 3.27 for spontaneous pregnancies after X-ray HSG employing oil-based contrast and an OR 1.13 for water-based contrasts [26]. However, when parallel studies are performed, the difference is much smaller [13]. Therefore, a possible publication bias could exist. Nevertheless, this increase in the probability of pregnancy represents one of the reasons supporting the use of this diagnostic technique in infertile women, even if it employs ionizing radiation and iodinated contrast and is rather uncomfortable [13].

However, ultrasound examinations using either a mixture of air bubbles in saline or sonographic contrasts media like SonoVue^TM^, or more recently ExEm^®^ foam are becoming first-line tubal patency tests [27]. Considering that HyFoSy is called to replace HSG, it is important to assess if there is any positive effect on pregnancy rates after HyFoSy. We have analyzed the effects of HyFoSy on pregnancy rate in infertile women with at least one fallopian tube patient and with a partner with a normal seminal test or suitable semen for IUI. 

We report that 26% of the women conceived spontaneously in the first 12 months after HyFoSy using ExEm^®^ foam. Tanaka and colleagues reported a pregnancy rate close to 50% in their study [14]. These data are more optimistic than our results. Our data are rather in agreement with data reported by Exacoustos and co-workers in 2015, who reported a 34.4% rate after HyFoSy [28]. Giugliano and colleagues assessed the therapeutic effect of the tubal patency test HyFoSy using bubble air in saline solution in an observational prospective study with 180 patients [18]. Their pregnancy rate was 22.2% within 6 months, with a rate of spontaneous abortions of 7.5%. These results are very similar to ours even if our proportion of spontaneous abortions is slightly higher [12%]. Previous reports have shown the safety profile of ExEm^®^ gel for both gametes and embryos [29]. Hence, the first-trimester miscarriage rate is similar to or less in our spontaneous pregnancies compared with other series of subfertile women [30]

Regarding conception time, our patients spent a median of 4 months [IQR 3.093–4.960] to achieve spontaneous pregnancy, with 18.9% of the women conceiving in the same month of the exam. Chunyan and colleagues and Van Schouboreck and co-workers also found the highest proportion of SC in the same cycle of the HyFoSy [19,31]. In fact, Chunyuan and colleagues observed that the spontaneous conception rate was significantly higher within the first 180 days after the procedure (19.4%) than later (6.3%) [19]. In fact, the cumulative conception rate we observed is higher than that observed in infertile couples that attempt to conceive within 12 months after diagnosis of infertility, rated as 14–20% [32,33]. Interestingly, looking at Figure 2, we observe a similar curve to other curves from women without infertility [34]. 

We found that 9 out of 201 [4.5%] women with bilateral occlusions at the HyFoSy examination got spontaneously pregnant. This could be explained by the probability of a false positive diagnosis for tubal occlusion inherent in the technique, which is as high as 5% [35]. This false occlusion could be due to either stromal edema, spasms of the tubes, mucus plugs in the ostium, or the presence of intrauterine blood clots.

The impact of basal characteristics of the women on the probability of achieving a spontaneous pregnancy has been studied in our series. Patients under 35 years with unexplained sterility had an OR of 2.22 [IC 95% 1.07–1.65] to achieve SC [*p* 0.033]. According to the sterility time, women with less than 18 months of infertility problems had an OR of 1.93 [1.34–2.81] *p* < 0.001 of becoming spontaneously pregnant. It is clear that, from the fertility point of view, these women constitute an optimal group. However, our data would support the concept that young women, with an unexplained and short time of sterility, should be ideal candidates to achieve post-HyFoSy SC. HyFoSy could be not only diagnostic but also therapeutic without assuming severe complications. It should be noted that a priori, from the reproductive point of view, these patients are those with the most favorable prognosis. Probably, tubal plugs might be present and they are removed with the pressure exerted by introducing the contrast. Additionally, these women had theoretically better ovarian reserve, and there is no male factor associated. Probably, HyFoSy “per se” would not have this effect in women with severe endometriosis, low or poor ovarian reserve, or cases with severe male factor. On the other hand, tubal disease might not cause complete occlusion. Damage may have been sufficient to hinder visualization of dispersing foam, but there may have been microscopic patency, and subsequently, patients may get pregnant.

### 4.3. Strengths and Limitations

The main strength of our study is that, to our knowledge, our study is one of the first large multicentric prospective studies assessing which factors are associated with spontaneous conception after the tubal patency test using the HyFoSy technique.

However, our study has limitations. First, there is no control group, so we cannot truly evaluate the potential “therapeutic effect” of HyFoSy. Second, our series is large, but we did not estimate sample size and statistical power. Third, some data were missing in a number of patients, but we did not exclude them.

## 5. Conclusions

In conclusion, our data confirm that spontaneous pregnancy can be achieved after the HyFoSy test for assessing tubal occlusion. On the other hand, HyFoSy seems well tolerated and detects 25% of women as having unilateral occlusion, which may guide management. In view of our data, and if stronger evidence studies would find similar results, it might even be possible to consider repeating the test in young women with short infertility time after a few months before submitting them to more complex assisted reproductive techniques. However, we acknowledge that further studies with stronger evidence are needed to continue investigating the possible therapeutic effects of HyFoSy. 

## Figures and Tables

**Figure 1 diagnostics-13-00504-f001:**
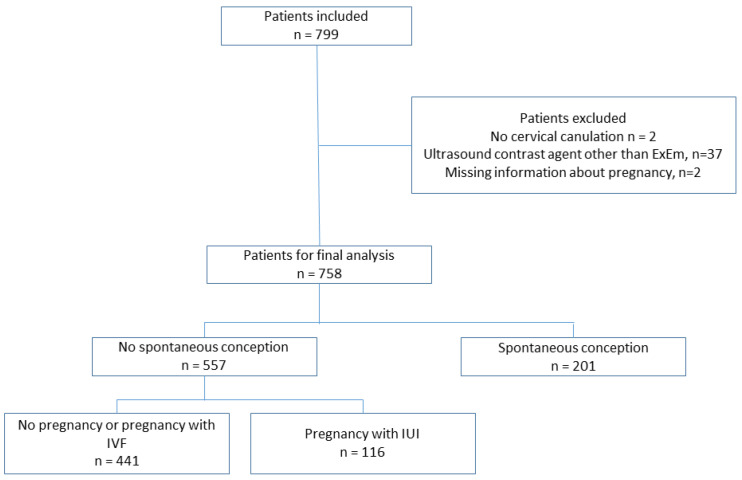
Flow-chart of patients included in the study.

**Figure 2 diagnostics-13-00504-f002:**
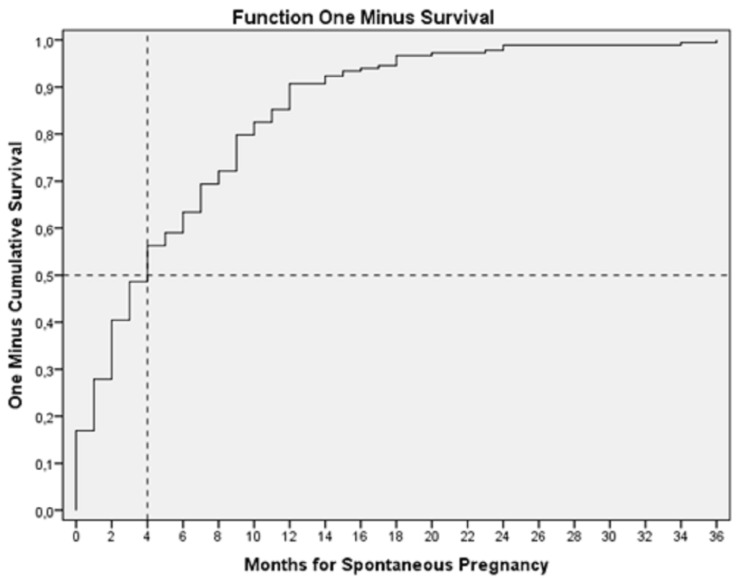
Kaplan-Meier graphic showing pregnancy rate after HyFoSy.

**Table 1 diagnostics-13-00504-t001:** Baseline characteristics and fertility parameters.

Characteristics		Total, Patients	No Spontaneous Conception (NSC)	Spontaneous Conception (SC)	*p*-Value
*n* (%)		758	557 (73.5)	201 (26.5)	
BMI (median, IQR)			22.9 (20.6–25.6)	23.0 (20.0–27.0)	N.S.
Age, years (median, IQR)			34 (31–36)	34 (30–36)	N.S.
Age < 35 years			324 (58.2%)	132 (65.6%)	0.0754
Time of sterility, months (median, IQR)			18 (12–24)	18 (12–24)	0.0034
Time of sterility < 18 months			306 (55.9%)	139 (69.1%)	0.0006
Sterility etiology	Unexplained * Ovulatory ** Mild or moderate male factor ** Tubal factor ** Others(endocrinology, uterine…) ** Mixed factor **	228/652 (35.0) 113/652 (17.3) 114/652 (17.5) 52/652 (8.0) 70/652 (10.7) 75/652 (11.5)	151/469 (32.2) 94/469 (20.0) 85/469 (18.1) 32/469 (6.8) 46/469 (9.8) 61/469 (13.0)	77/183 (42.1) 19/183 (10.4) 29/183 (15.8) 20/183 (10.9) 24/183 (13.1) 14/183 (7.7)	* 0.017 ** N.S.

Data shown as *n* (% of total), except otherwise indicated. The inclusion of a denominator indicates missing data for that particular characteristic. N.S. Not significant. BMI: body mass index. * means *p* = 0.017. ** means N.S., not statistical significance.

**Table 2 diagnostics-13-00504-t002:** HyFoSy parameters.

Characteristic		Total	No Spontaneous Conception	Spontaneous Conception	*p*-Value
		758	557 (73.5)	201 (26.5)	
Contrast volume (CC) (median, IQR)		5.0 (3.0–7.0)	5.0 (3.0–7.0)	5.0 (3.0–8.0)	0.314
Tubal patency	Both tubes patent One tube patent Bilateral occlusion Not valuable	576/755 (76.3) 146/755 (19.3) 32/755 (4.2) 1/755 (0.1)	422/554 (76.2) 108/554 (19.5) 23/554 (4.2) 1/554 (0.2)	154 (76.6) 38 (18.9) 9 (4.5) 0 (0.0)	0.934
PAIN (VAS score)	Mild (0–3) Moderate (4–7) Severe ( > 8)	456/712 (64.0) 227/712 (31.9) 29/712 (4.1)	331/521 (63.5) 169/521 (32.4) 21/521 (4.0)	125/191 (65.4) 58/191 (30.4) 8/191 (4.2)	0.871

Data shown as *n* (%), except otherwise indicated. The inclusion of a denominator indicates missing data for that characteristic.

**Table 3 diagnostics-13-00504-t003:** Multivariable analysis of the risk of spontaneous pregnancy after HyFoSy examination.

Initial Maximal Model	Final Estimation Model	Variables Conserved in the Model	*p*-Value	OR (95% CI)
Spontaneous pregnancy = interaction (cause of sterility + age categorized) + time of sterility categorized	Spontaneous pregnancy = interaction (cause of sterility + age categorized) + time of sterility categorized	Time of sterility categorized	<0.001	1.93 (1.34–2.81)
		Age categorized	0.216	
		Unknown sterility	0.830	
		Unknown sterility in women aged under 35 years	0.033	2.22 (1.07–4.65)

Cause of sterility: unknown (1) vs. other sterility factors (0); Categorized age: <35 (1) vs. ≥35 (0); Categorized time of sterility: ≤18 months (1) vs. >18 months (0).

**Table 4 diagnostics-13-00504-t004:** Perinatal outcomes HyFoSy and subsequent intrauterine inseminations vs. HyFoSy alone.

Characteristics		Total, Patients	HyFoSy + IUI	HyFoSy	*p*-Value
Pregnancy	Term delivery Preterm delivery miscarriage	220/267 (82.4) 20/267 (7.5) 27/267 (10.1)	76/92 (82.6) 10/92 (10.9) 6/92 (6.5)	144/175 (82.3) 10/175 (5.7) 21/175 (12.0)	0.140
Evolution of pregnancy	Ongoing pregnancy miscarriage	240/267 (89.9) 27/267 (10.1)	86/92 (93.5) 6/92 (6.5)	154/175 (88.0) 21/175 (12.0)	0.158
Neonatal weight, grams (median, IQR)		3176 (2850–3430)	2998 (2656–3485)	3205 (3016–3395)	0.034 *
Fetal malformations	Yes No	12/192 (6.2) 180/192 (93.8)	7/73 (9.6) 66/73 (93.8)	5/119 (4.2) 114/119 (95.8)	0.217

Data shown as *n* (% of total), except otherwise indicated. The inclusion of a denominator indicates missing data for that particular characteristic. * Statistically significant differences.

## Data Availability

Data are available upon reasonable request.

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
