# Peer review of "Factors Associated with a Post-Procedure Spontaneous Pregnancy after a Hysterosapingo-Foam-Sonography (HyFoSy): Results from a Multicenter Observational Study"

_diagnostics, 2023, doi:10.3390/diagnostics13030504_

Round 1
Reviewer 1 Report
I appreciate the opportunity to review the manuscript entitled “Factors associated to post-procedure spontaneous pregnancy after Hysterosapingo-foam-sonography (HyFoSy). Results from a multicenter observational study” submitted to the journal Diagnostic.
The authors investigated the predictive value of the different variables on the chance to achieve pregnancy after HyFoSy. Taking into account that infertility is a health, medical and economic problem, the idea of this research is important.
Reviewer Comments:
1. The results observed in this research are not new however the paper has no scientific soundness.
2. The results observed in this research do not give new and important data in the field of infertility.
.
Taking into account the lack of scientific soundness of the above-mentioned paper, my opinion is that this submission does not meet the criteria to be published in the journal Diagnostic.
Author Response
I appreciate the opportunity to review the manuscript entitled “Factors associated to post-procedure spontaneous pregnancy after Hysterosapingo-foam-sonography (HyFoSy). Results from a multicenter observational study” submitted to the journal Diagnostic. The authors investigated the predictive value of the different variables on the chance to achieve pregnancy after HyFoSy. Taking into account that infertility is a health, medical and economic problem, the idea of this research is important.
- Question: The results observed in this research are not new however the paper has no scientific soundness.
- Answer: With all respect to the reviewer, we do think we provide new data, such as what factors can be associated to spontaneous pregnancy after HyFoSy
- Question: The results observed in this research do not give new and important data in the field of infertility.
- Answer: With all respect. See previous answer
Reviewer 2 Report
Congratulations for the authors, paper is well-written. The conception, the design of the study, the statistics and the presentation is very good.
There are some comments for revision:
- this phenomenon is called "tubal flushing effect" in the literature. This name should be mentioned and should add to the keywords, as well
- there are other out-patient tubal patency test, that should be mentioned (via office hysteroscopy)
- pain score should be analyzed in patent vs blocked tube group. Blockage of tube can increase the intrauterine pressure and that is how the pain.
Considering these small changes, the paper is adequate for publication
Author Response
Congratulations for the authors, paper is well-written. The conception, the design of the study, the statistics and the presentation is very good. There are some comments for revision:
- Question: this phenomenon is called "tubal flushing effect" in the literature. This name should be mentioned and should add to the keywords, as well
- Answer: thanks for this comment. We added keywords and also mentioned this effect in the text (See lines 74 and 77)
- Question: there are other out-patient tubal patency test, that should be mentioned (via office hysteroscopy)
- Answer: Reviewer is right. We mention this in the Introduction. A reference has been added. See lines 59-62
- Question: pain score should be analyzed in patent vs blocked tube group. Blockage of tube can increase the intrauterine pressure and that is how the pain.
- Answer: We already performed this analysis in our previous paper (Engels V, Medina M, Antolín E, Ros C, Amaro A, De-Guirior C, Manzour N, Sotillo L, De la Cuesta R, Rodríguez R, San-Frutos L, Peralta S, Martin-Martínez A, Alcázar JL. Feasibility, tolerability, and safety of hysterosalpingo-foam sonography (HyFoSy). multicenter, prospective Spanish study. J Gynecol Obstet Hum Reprod 2021;50:102004). For this reason, we did not provide this data in the present manuscript. No change made in the manuscript.
Reviewer 3 Report
The submission is a large sample size study of differences among women who successfully conceive after HyFoSy and those that do not. There are some limitations to design and interpretation, but with refinement, this submission would be a useful addition to the literature.
The number one things the authors need to better address is an inherent bias to the literature for their submission through the healthy user effect. E.G. for Dreyer, only 8% of patients having HSG had tubal disease as a source for subfertility. How much is the HyFoSy truly improving fertility if the tubes are already normal? Moreover, would HyFoSy improve endometriosis, low sperm counts, ovarian failure, anovulation, aneuploidy, submucosal fibroids, etc.? Would it even cause ciliary regeneration for those with intraluminal tubal damage? If most patients don’t have tubal related disease and HyFoSy isn’t addressing the pathology, why does the HyFoSy work other than observational/interventional bias or the healthy user (not trying to conceive, and now they are)? The core limitation to this paper is the implication that HyFoSy is improving fertility when directly doing so is biologically dubious.
1. In the abstract, ExEm Foam has the 2nd E capitalized as part of the trademarked brand.
2. The authors note a significant increase in spontaneous pregnancy rates after HSG, but cite the wrong article. The 2017 NEJM article compares oil and water based HSG; the proper article from the same authors should be their 2019 RBMO article, comparing HSG to no HSG on pregnancy rates (and showing a hazard ratio of 1.4-1.48, which is arguable as to how significant it is of a boost, particularly in patients with lower spontaneous pregnancy rates)
3. The authors are well within their rights to use spontaneous conception as part of a case-control approach. However, if arguing that HyFoSy meaningfully improves pregnancy rates, the control group should be those not having HyFoSy, which was not done. This is a meaningful limitation to the paper.
4. The authors should discuss/note how in Figure 2, the time to conception heavily parallels that of those without infertility (e.g. Gnoth and other life-table analyses for spontaneous conception).
5. As feedback, Table 2 is wonderful—the data on the probability of unilateral/bilateral occlusion, as well as pain experience is underaddressed in the literature for large sample size publications.
6. For the first discussion point, the authors need to address spontaneous pregnancy rates without HyFoSy for the general population. 26.5% of patients with a year to a year and a half of subfertility conceiving is a fairly normal statistic without evaluation.
7. In referencing in 4.2 the gap between oil and water based contrast, when parallel studies are performed, the difference is much smaller. (E.G. Dreyer’s publications). Accordingly, publication bias for the Cochrane summary should be acknowledged.
8. When addressing reasons patients with tubal occlusion on HyFoSy subsequently spontaneously conceived, the authors should acknowledge an additional possibility—HyFoSy may have correctly diagnosed tubal damage. However, tubal disease need not be complete occlusion or complete patency. Damage may have been sufficient to hinder visualization of dispersing foam, but there may have been microscopic patency and subsequently the patients got lucky.
9. The conclusion is weak, where without a proper control group, saying that spontaneous conception can occur after HyFoSy doesn’t really add to the literature. More powerfully, stating that HyFoSy seems well tolerated and detects 25% of women as having unilateral occlusion which may guide management.
The authors add a lot of insight despite limitations to design. I hope to see this submission in print after refinement.
Author Response
The submission is a large sample size study of differences among women who successfully conceive after HyFoSy and those that do not. There are some limitations to design and interpretation, but with refinement, this submission would be a useful addition to the literature.
- Question: The number one things the authors need to better address is an inherent bias to the literature for their submission through the healthy user effect. E.G. for Dreyer, only 8% of patients having HSG had tubal disease as a source for subfertility. How much is the HyFoSy truly improving fertility if the tubes are already normal? Moreover, would HyFoSy improve endometriosis, low sperm counts, ovarian failure, anovulation, aneuploidy, submucosal fibroids, etc.? Would it even cause ciliary regeneration for those with intraluminal tubal damage? If most patients don’t have tubal related disease and HyFoSy isn’t addressing the pathology, why does the HyFoSy work other than observational/interventional bias or the healthy user (not trying to conceive, and now they are)? The core limitation to this paper is the implication that HyFoSy is improving fertility when directly doing so is biologically dubious.
- Answer: We fully agree with the reviewer. In fact, we comment this in the revised version, in the Discussion section.
- Question: In the abstract, ExEm Foam has the 2ndE capitalized as part of the trademarked brand.
- Answer: Corrected
- Question: The authors note a significant increase in spontaneous pregnancy rates after HSG, but cite the wrong article. The 2017 NEJM article compares oil and water based HSG; the proper article from the same authors should be their 2019 RBMO article, comparing HSG to no HSG on pregnancy rates (and showing a hazard ratio of 1.4-1.48, which is arguable as to how significant it is of a boost, particularly in patients with lower spontaneous pregnancy rates)
- Answer: Thanks for this comment. Reference amended
- Question: The authors are well within their rights to use spontaneous conception as part of a case-control approach. However, if arguing that HyFoSy meaningfully improves pregnancy rates, the control group should be those not having HyFoSy, which was not done. This is a meaningful limitation to the paper.
- Answer: We fully agree. In fact, we stress this point as a limitation. See line 336. No change made in the revised version
- Question: The authors should discuss/note how in Figure 2, the time to conception heavily parallels that of those without infertility (e.g. Gnoth and other life-table analyses for spontaneous conception).
- Answer: We comment on this in the Discussion. New reference added
- Question: As feedback, Table 2 is wonderful—the data on the probability of unilateral/bilateral occlusion, as well as pain experience is under-addressed in the literature for large sample size publications.
- Answer: Thanks for this comment.
- Question: For the first discussion point, the authors need to address spontaneous pregnancy rates without HyFoSy for the general population. 26.5% of patients with a year to a year and a half of subfertility conceiving is a fairly normal statistic without evaluation.
- Answer: Discussion modified accordingly to suggestion. New references added
- Question: In referencing in 4.2 the gap between oil and water based contrast, when parallel studies are performed, the difference is much smaller. (E.G. Dreyer’s publications). Accordingly, publication bias for the Cochrane summary should be acknowledged.
- Answer: Thanks for this comment. Text modified accordingly to suggestion.
- Question: When addressing reasons patients with tubal occlusion on HyFoSy subsequently spontaneously conceived, the authors should acknowledge an additional possibility—HyFoSy may have correctly diagnosed tubal damage. However, tubal disease need not be complete occlusion or complete patency. Damage may have been sufficient to hinder visualization of dispersing foam, but there may have been microscopic patency and subsequently the patients got lucky.
- Answer: Thanks for this comment. Conclusion modified accordingly to suggestion.
- Question: The conclusion is weak, where without a proper control group, saying that spontaneous conception can occur after HyFoSy doesn’t really add to the literature. More powerfully, stating that HyFoSy seems well tolerated and detects 25% of women as having unilateral occlusion which may guide management.
- Answer: Thanks for this comment. Conclusion modified accordingly to suggestion.
Reviewer 4 Report
The authors demonstrated to evaluate the post procedure rate of spontaneous pregnancy among infertile women submitted to HyFoSy exam with Exem-foam and the factors associated to this. I think the study was very well designed and the interpretation of the results is appropriate. I would like to address only the minor points in the following.
P3L83: On the other, hand → On the other hand,
P7L205: The authors mentioned that “we observed that 205 patients aged below 35 years tended to get pregnant after HyFoSy more frequently that 206 those aged over 35 years old (Table 1).” However, since there is no significant difference in age, this description might be better revised.
P7L218: The authors mentioned that “Median duration of infertility was 18 months (IQR 12-24) in both groups.” However, Table 1 shows a p-value of 0.0034. Is table 1 incorrect?
Table 1:Sterility etiology is shifted.
OP7L217: p=0-017→p=0.017
P11L318: I would definitely encourage the authors to include a discussion of the reasons for the higher pregnancy rates in women of younger age, shorter duration of infertility, and unexplained infertility.
Author Response
See pdf attached

Round 2
Reviewer 1 Report
I appreciate the opportunity to review the revised version of the manuscript entitled “Factors associated to post-procedure spontaneous pregnancy after Hysterosapingo-foam-sonography (HyFoSy). Results from a multicenter observational study” submitted to the journal Diagnostic.
The authors investigated the predictive value of the different variables on the chance to achieve pregnancy after HyFoSy. Taking into account that infertility is a health, medical and economic problem, the idea of this research is important.
I think that the manuscript has been improved and only now it meets the criteria to be published in the journal Diagnostic.
Author Response
Dear Reviewer,
Thanks for your comments. We appreciate
No change made in the manuscript.
Reviewer 3 Report
Though requiring a bit more editing for grammar in the new portions, the authors have made reasonable and appropriate edits in general. A slight modification is that for reference 7, that citation actually did not perform selective hysteroscopic chromopertubation (though Torok did in 2012; however, that was actually not office based despite suggestion of such in the title and other points in the manuscript.) For a review of hysteroscopic approaches, please consider Vitale, et. al., in JMIG, 2021. However, otherwise the paper seems notably improved.
Author Response
Dear Reviewer
Thanks for your comment.
We agree. We have modified the reference #7 and changed by Vitale et al. No other changes made in the manuscript.
Sincerely yours,
Juan Luis Alcázar